# Sex-Related Analgesic Effects of Opioid-Based Anesthesia and Low-Opioid Anesthesia with Non-Opioid Postoperative Analgesia—A Clinical Observational Study

**DOI:** 10.3390/jcm14072163

**Published:** 2025-03-22

**Authors:** Urszula Kosciuczuk, Agnieszka Kossakowska, Marcin Talalaj, Katarzyna Grabowska, Marta Pryzmont

**Affiliations:** Department of Anesthesiology and Intensive Therapy, Medical University of Bialystok, Kilinskiego 1 Street, 15-276 Bialystok, Poland; kossakowska-agnieszka@wp.pl (A.K.); marcinmt.mt@gmail.com (M.T.); k-grabowska94@wp.pl (K.G.); pryzmontmarta@wp.pl (M.P.)

**Keywords:** analgesia, opioid, pain, trajectory, women

## Abstract

**Background/Objectives:** Sex is a crucial factor in modulating the perioperative aspects of anesthesia. A growing number of studies demonstrate that women and men experience pain differently and respond differentially to analgesics. **Methods:** This study evaluated the analgesic trajectory of low-opioid anesthesia (LOA) and opioid-based anesthesia (OBA) in women and men after laparoscopic cholecystectomy. The primary objective was to assess pain intensity at various time intervals after surgery (0–2, 2–6, 6–12, and 12–24 h) using the Numerical Rating Scale (NRS). The secondary objective was to assess the difference in mean pain intensity on the first postoperative day between the women and men. **Results:** The mean pain intensity did not differ significantly for men using LOA and OBA, but the value was significantly lower in the LOA group for women (*p* = 0.0002). The analgesic trajectory in women and men undergoing LOA presented a negative trend, but the pain intensity at 0–2, 2–6, 6–12, and 12–24 h in women was statistically lower than that of the OBA group (*p* = 0.01, *p* = 0.008, *p* = 0.002, and *p* = 0.001). Total fentanyl doses of 0.3 mg (sensitivity 44%, specificity 76%, AUC 0.55) and 0.35 mg achieved a mean NRS of <2 for pain intensity in the female and male OBA groups (sensitivity 33%, specificity 100%, AUC 0.53). **Conclusions:** A model combining low-opioid anesthesia and non-opioid postoperative analgesia presents a favorable therapeutic effect for women. OBA does not provide proper analgesic effects after laparoscopic cholecystectomy.

## 1. Introduction

Opioid analgesics are an important pharmacological component of general anesthesia and have been widely used during the perioperative period for several decades. In 1962, opioid drugs were introduced in the practice of anesthesiology along with volatile anesthetics, which became the foundation of multimodal anesthesia [1,2,3,4]. General anesthesia consists of three crucial elements: unconsciousness, immobility, and the suppression of responses from the autonomic nervous system. The unique properties of opioids are inhibitory effects on nociception during anesthesia, analgesic effects in the postoperative period, and the suppression of sympathetic responses during anesthesia. Opioids are primary and multipotent antinociceptive agents that affect all points of the anesthesia triangle with a synergistic pharmacokinetics effect with intravenous and volatile anesthetics as well as benzenediazepines. The pharmacological effects of opioids reduce the adverse reaction to the various anesthetic activities, including tracheal intubation, and reduce the intensity of surgical stress. While the mechanisms of opioids affect the basic elements of general anesthesia, they are not ideal substances in terms of analgesia and safety due to their side effects, requiring constant efforts to limit their use. Deep general anesthesia based on the use of opioid analgesia is contraindicated in advanced cardiovascular disease and the use of this group of drugs should be considered [4,5,6,7,8].

In 1993, Friedberg reported on general anesthesia using an opioid-free combination of ketamine and propofol. While opioid-sparing methods—opioid-free anesthesia (OFA) and low-opioid anesthesia (LOA)—are a new phenomenon, the shift from opioid-based anesthesia (OBA) and postoperative analgesia as the sole analgesic component to balanced anesthesia and multimodal analgesia with opioids and coanalgesics is a logical and natural progression. Multimodal analgesic therapy is based on the theory of combining opioid analgesics, non-opioid analgesics, and coanalgesics to achieve benefits without the risk of addiction. A multimodal analgesic scheme can reduce pain and opioid consumption [9,10,11]. The number of publications and scientific interest in the topic of opioid-sparing analgesia has remained stable for several years, just as the number of publications addressing opioids in the therapeutic process of perioperative care and postoperative safety associated with their pre-drug use has grown very rapidly. Despite the evolution of analgesic strategies, the goal is always to optimize drug selection to provide optimal perioperative analgesia with long-term benefits and improved recovery trajectory [12,13,14].

The current understanding of opioid-sparing anesthesia and analgesia is a perioperative care strategy that maximizes non-opioid methods and reserves the use of opioids for severe acute pain that cannot be relieved by other methods. Conversely, OFA is an absolute avoidance of opioids in the intraoperative period. This includes the use of a wide range of medications to reduce the adverse effects of perioperative opioid use, such as hyperalgesia, respiratory depression, nausea, vomiting, itching, and a potential reduction in immunosuppression [15,16,17,18]. 

The predominant problem in implementing opioid-sparing methods is that opioids are the most effective pharmacologic agent used in many clinical situations, and single elements of multimodal analgesia are ineffective alone [19,20]. LOA and OFA are not always seen as preferential methods, as side effects (e.g., bradycardia and hypoxemia) occurred in 78% of patients after dexmedetomidine and in 68% after opioids. Meanwhile, hallucinations, prolonged paralysis with magnesium, bradycardia connected with beta-blockers, hypoglycemia, and hypotension are associated with ketamine use [21,22,23,24,25].

From a patient safety and perioperative comfort perspective, the expectations for opioid-sparing anesthesia and analgesia are as follows: reductions in nausea and vomiting, slowed gastrointestinal peristalsis and constipation, sedation, urinary retention, respiratory depression, delirium, and chest wall rigidity. Other expectations of postoperative analgesia include avoiding post-opioid hyperalgesia and increased opioid drug administration, as well as reductions in surgical wound hypersensitivity, chronic pain, and cancer recurrence [24,25,26,27,28].

Patients with chronic preoperative pain (especially on opioid therapy), obstructive sleep apnea syndrome, and obesity, as well as elderly patients, are expected to obtain the most benefits from LOA and OFA [10,24]. These methods apply to anesthesia for bariatric surgery, oncologic surgery, orthopedic surgery, gynecologic surgery, head and neck surgery, and thoracoscopy, and meet ERAS expectations [29,30,31,32].

There is no universal pharmacological opioid-sparing scheme. The most common coanalgesics include ketamine (an NMDA receptor antagonist), lidocaine (an amide group of local anesthetics that cause a sodium channel blockade), dexmedetomidine (a highly selective alpha 2 adrenoreceptor agonist with sedative and anxiolytic actions), magnesium (a calcium channel and NMDA receptor blockade), and esmolol (a short-acting beta blocker that causes sympatholytic effects) [33,34,35,36].

A multimodal view of opioid-sparing methods involves a complex pharmacological system of antinociceptive substances affecting multiple levels of nociceptive transmission with additional sedative and anxiolytic effects. This represents an innovative approach to pain management that prioritizes patient safety, reduces the risks associated with opioid use, and seeks to enhance recovery outcomes. These methods reduce postoperative complications during the recovery period (nausea and vomiting) and are particularly beneficial for patients who are most susceptible to opioid-related complications [37,38,39,40,41,42].

Nociception and analgesic perception have been suggested to be sex-dependent. Although women have a higher pain threshold, they present a wait-and-see strategy of analgesic use, with longer periods of use, more frequent reports of chronic pain, atypical clinical manifestations, and neuropathic pain. Combined with the laparoscopic method of surgery and young age, the sex factor is an important risk factor in postoperative nausea and vomiting syndrome [43,44,45,46].

This study aimed to evaluate the analgesic trajectory of low-opioid anesthesia (LOA) and opioid-based multimodal anesthesia (OBA) in a group of women and men after laparoscopic cholecystectomy. We hypothesized that limiting the use of opioid drugs through LOA and a non-opioid postoperative analgesia regimen would provide an analgesic profile in groups of women and men comparable to OBA. The primary objective was to assess pain intensity at time intervals after surgery (0–2, 2–6, 6–12, and 12–24 h). The secondary objective was to assess the mean pain intensity on the first postoperative day between the female and male groups.

## 2. Materials and Methods

This study was approved by the Bioethics Committee of the Medical University of Bialystok, No. R-I-002/105/2019, and included adult patients scheduled for elective laparoscopic cholecystectomy. The exclusion criteria included a lack of consent to participate in the study; an ASA III or IV health status; a history of cancer and chronic pain; liver cirrhosis; epilepsy; addiction to drugs, alcohol, or psychoactive substances; and a history of postoperative nausea and vomiting, allergic reactions, or contraindications to analgesics. This was a non-blinded cohort study, and all of the patients were informed of the purpose of the intended study and how it was to be conducted, and they provided written consent.

This study included 80 adult patients undergoing laparoscopic cholecystectomy under combined general anesthesia with intravenous anesthesia induction (propofol) and the volatile maintenance of anesthesia (sevoflurane) using cisatracurium as a skeletal muscle relaxant. The subjects were divided into two groups according to the analgesic method used:

Study group: 40 patients who received low-opioid anesthesia and analgesia (LOA);

Control group: 40 patients who received opioid-based anesthesia and analgesia (OBA).

In the LOA method only one dose of fentanyl was used in the induction of anesthesia, and additional elements of analgesia were composed of paracetamol, magnesium, lidocaine, and ketamine in bolus doses, and then an infusion of lidocaine and ketamine in the maintenance phase. In the OBA method repeated doses of fentanyl were administered—first dose in the induction of anesthesia and additional doses in the maintenance phase. The postoperative scheme of analgesia was the same in all patients—1 g of paracetamol and 1 g of metamizole intravenously every 6 h—and it was characteristic non-opioid analgesia.

The designs of LOA and OBA are presented in Table 1. 

Both methods of general anesthesia were considered to be equivalent in analgesic potency and we initially recruited 40 patients to the LOA group, and then recruited 40 patients to the OBA group. The inclusion criteria for the LOA and OBA groups were the same as for recruitment in this study. The study assumption was that both methods in patients with preoperative states of ASA 1 and 2 have no contraindications to anesthesia with these methods.

Pain severity was assessed at time intervals after surgery, including the first 2 h and between 2 and 6, 6 and 12, and 12 and 24 h. A numerical scale was used, according to which the number “0” indicates the absence of pain, and the number “10” expresses the most severe level of pain imaginable.

## 3. Statistical Analysis

Data are presented as the median, minimum–maximum range, and 25–75th percentile range (IQR), or as counts (n) with proportions (%). The analysis was performed on groups of women and men. The Shapiro–Wilk test was used to check the normality of the variable distribution. For normal distribution values, parametric tests to compare independent variables and independent groups were used. Non-parametric tests were used to represent correlations between variables with non-normal distribution and between groups. The Mann–Whitney U-test was used to compare two independent groups. All calculations were performed using Statistica 14 (Poland, TIBCO, Cracow, Poland), and *p* < 0.05 was the level of significance. This was a prospective study; therefore, the sample size was 40 subjects in the experimental group. We checked the sample size calculation based on the mean NRS value, with an alpha of 0.05 and a power goal of 0.8, calculating a minimum group size of 21.

## 4. Results

In total, 80 patients who fulfilled the inclusion criteria participated in this study. One patient from the LOA group and three patients from the OBA group discontinued their participation and were excluded from the analysis. Finally, 39 patients were in the LOA group, and 37 patients were in the OBA group. The demographic and clinical characteristics of the patients were comparable between groups. There was no statistically significant difference in the amount of fentanyl used between women and men in the OBA group. Indeed, the durations of anesthesia and surgery were similar between the two groups. The characteristics of the study and control groups are presented in Table 2.

The mean pain intensity on postoperative day 1 was comparable between men and women in the LOA group. It was 2.25 (min–max 0–4, IQR 2.0–3.0) for men and 1.62 (min–max 0–7.0, IQR 1.0–2.5) for women. In contrast, higher values of mean pain intensity were recorded in the OBA group for women—4.12 (min–max 1.5–7.25, IQR 3–4.75) compared with men—3.0 (min–max 1.0–6.0, IQR 1.5–5.0). There was no statistically significant difference in mean pain intensity on postoperative day 1 between the LOA and OBA groups for men, but there was a significantly lower mean pain intensity in the LOA group for women (*p* = 0.0002; Figure 1).

The analgesic trajectory in women undergoing LOA presented a negative trend, and the pain intensity values recorded in time intervals of 0–2 h, 2–6 h, 6–12 h, and 12–24 h were statistically lower than those for the OBA group (*p* = 0.01, *p* = 0.008, *p* = 0.002, and *p* = 0.001, respectively) (Figure 2). A negative trend was also observed for men in the LOA group. A positive slope was recorded for men in the OBA group, and NRS values did not differ significantly in the 0–2, 2–6, and 6–12 intervals between the groups (*p* = 0.7, *p* = 0.8, and *p* = 0.07, respectively). NRS values in the 12–24 h interval were significantly lower among men in the LOA group (*p* = 0.03; Figure 2). 

The NRS pain intensity values for each time interval are presented in Table 3.

## 5. Discussion

Our study aimed to compare the effects of LOA and OBA schemes used on women and men undergoing laparoscopic cholecystectomy. The proposed model of general anesthesia based on lidocaine, ketamine, and magnesium created a favorable analgesic profile (i.e., with a negative slope) among both women and men. However, we observed significantly low NRS values only in the female LOA group compared with the OBA group. In the male LOA group, only the pain intensity at the end of postoperative day 1 (12–24 h) was significantly lower than in the OBA group. OBA did not guarantee an effective analgesic profile in the female and male groups. In the female group, the analgesic trajectory presented a neutral slope, while in the male group, a positive slope was observed. The analysis indicated that to achieve comparable postoperative analgesia in the OBA group, the total dose of fentanyl should be 0.3 mg for women and 0.35 mg for men.

Various models of anesthesia for laparoscopic cholecystectomy surgeries based on limiting the use of opioid drugs have been described in the literature [47,48,49,50]. Vishnuraj et al. demonstrated that multimodal general-anesthesia-based ketamine and dexmedetomidine reduced the need for additional analgesia with fentanyl up to 2 h after surgery and reduced nausea and vomiting. In addition, there was no significant difference in opioid drug administration postoperatively. However, the time to administration of the first dose of fentanyl in the postoperative period was longer in the OFA group—about 17 min compared with 11 min for the OBA group. The average pain intensity was comparable at 6, 12, and 24 h after surgery in both groups [51].

Furthermore, the multimodal anesthesia model for laparoscopic cholecystectomy using dexmedetomidine and lidocaine showed a significant reduction in pain intensity and the need for additional postoperative analgesia. At the same time, there were more episodes of hypertension in the OFA group, and fentanyl consumption in the Patient-Controlled Analgesia group was significantly lower up to 2 h after surgery (75 µg vs. 120 µg) [52].

The practical implementation of OFA or LOA suggested in Option 1 is appropriate for minor surgical procedures (including one-day surgeries) with minimal to moderate postoperative pain. Locoregional analgesia and magnesium sulfate in an initial bolus of 30–50 mg kg^−1^ and a subsequent infusion of 40–60 mg kg^−1^ h^−1^ are preferred. Option 2 is appropriate for moderate surgical procedures with moderate to severe pain for opioid users. In this situation, in addition to regional analgesia, ketamine is used at a dose of 0.5 mg kg^−1^ upon induction and 0.25 mg kg^−1^ h^−1^ for maintenance. Ketamine is contraindicated in hypertension, eclampsia, severe heart failure, and stroke. Option 3A is appropriate for major surgeries lasting less than 2 h, invasive surgery with moderate to severe pain, or procedures that require an additional technique to stabilize hemodynamics. Dexmedetomidine is administered at a dose of 0.2–0.8 µg kg^−1^. Option 3B is preferred for major surgical procedures lasting longer than 2 h. For additional hemodynamic stability, clonidine is used at a dose of 1–4 µg kg^−1^. As the first line, paracetamol, ibuprofen, and locoregional analgesia can be used as additional analgesic treatments for acute pain, with one opioid as a rescue medication. The second line is ketamine for neuropathic pain or opioid resistance or nefopam to reduce opioid-induced sedation. The third line is gabapentin and/or clonidine [1].

The PROSPECT (procedure-specific postoperative pain management) study group presented recommendations for perioperative pain management after laparoscopic surgery. In general, they recommended basic analgesic techniques: paracetamol + NSAIDs or cyclooxygenase-2 inhibitor + topical local anesthesia at the surgical site. As Grade A analgesics, paracetamol and NSAIDs should be started before or during surgery with dexamethasone. Opioids should only be used for rescue analgesia and are classified as Grade B. Gabapentanoids, intraperitoneal local anesthesia, and transverse abdominal plane blocks are not usually recommended and are considered Grade D. Surgically, low-pressure peritoneal emphysema, postoperative saline lavage, and the aspiration of peritoneal emphysema are suggested as Grade A strategies. Single-port incision techniques and heated and humidified CO_2_ insufflation are not recommended for pain relief (Grade A) [53].

Sex is a crucial factor modulating pain sensation, with a growing number of studies demonstrating that men and women experience pain differently and respond differentially to analgesics [47]. Therefore, this is an important factor in determining the course of general anesthesia and postoperative safety. 

Biological, cultural, psychological, and social factors all contribute to detecting, reporting, and treating pain, as well as the effectiveness of analgesia. Sex has been suggested as an independent factor in perioperative status, as well as in the speed of recovery from general anesthesia and sensitivity to analgesics. The reason for many physiological and functional distinctions is hormonal changes associated with cyclic trends in sex hormones, such as estrogen, progesterone, folliculotropic hormone, and luteinizing hormone. Women tend to have a higher body fat composition (by up to 10%) and lower muscle mass (by up to 10%) than men. Moreover, women have a lower total body water composition (up to 20%), and the extracellular fluid volume and plasma sodium levels change during the menstrual cycle. Estrogen and progesterone affect plasma volume and capillary fluid dynamics through renal sodium absorption and colloid osmotic pressure. The plasma sodium concentration increases in the mid-follicular and ovulatory phases and decreases in the luteal phase. Female sex hormones and androgens can cause changes in the cardiovascular system, including the vasodilation of blood vessels due to nitric oxide secretion, lowering blood pressure. In general, women have a lower heart mass, better diastolic function with a higher left ventricular ejection fraction and stroke volume, and a resting heart rate with a shorter heart cycle length. The lung anatomy also differs, with a lower lung volume, expiratory flow rate, and lung diffusion area than in men. Women also have a lower ventilator response to hypercapnia and hypoxia. All these differences lead to implementation considerations in anesthesiological aspects. Most important in the perioperative period is the volume of distribution, which is reduced for muscle relaxants and increased for diazepam, midazolam, and propofol. The Minimal Alveolar Concentration values for volatile anesthetic agents and postoperative nausea and vomiting syndrome are also closely related to the female sex. Women are most likely to report postoperative pain, sore throats, postoperative headaches, and worse postoperative outcomes [54,55,56,57,58,59]. Surprisingly, the sex factor also determines sensitivity to opioid analgesics. Women demonstrate greater sensitivity to the impacts of opioids through mi and kappa opioid receptors. However, the differences in the effects of non-opioid drugs remain unclear, and the conclusions of many studies are divergent [47].

Our results are consistent with data from other publications. Yeh et al. found that after multimodal general anesthesia, men required significantly higher doses of morphine after hepatic cancer surgery. Furthermore, after cardiac surgery, significantly higher opioid drug requirements were observed among men—a 139 mg oral morphine equivalent dose (OMED) vs. 180 mg. Conversely, the total morphine dose used from day 1 to 5 was significantly lower in women. However, there was no indication of a relationship between sex and the use of non-opioid medications and pain intensity within 48 h after extubation [56]. Acute postoperative pain is more prevalent among men, whereas chronic neuropathic pain is more frequently reported among women [45,49]. Estrogen has pro-nociceptive and pro-inflammatory effects due to macrophages, natural killer cells, dendric cells, and T lymphocytes. In addition, TLR (Toll-Like Receptor 4) expression increases in microglia in the periaqueductal gray, resulting in increased sensitivity in women [49].

Yang et al. identified preoperative predictors of poor acute postoperative pain control. The significant preoperative predictors were younger age (OR 1.18), female sex (OR 1.29), smoking (OR 1.33), a history of depressive symptoms (OR 1.71), a history of anxiety (OR 1.22), sleep difficulties (OR 2.32), a higher BMI (OR 1.02), preoperative pain (OR 1.21), and the use of preoperative analgesia (OR 1.54) [59,60,61].

This study has several limitations. The number of patients was too small to draw definitive conclusions, especially when subdivided by sex. This is a subanalysis of a study which primarily compared the effects of LOA and OBA. Based on the preliminary results, this study should be supplemented with additional information about lifestyle (e.g., nicotine use), coexisting psychiatric disorders, and preoperative analgesic therapy, as well as non-invasive monitoring of nociception. Secondarily, we checked the sample size calculation post-hoc based on the mean NRS value, with an alpha of 0.05 and a power goal of 0.8. The method of power calculation conducted a priori may improve the statistical significance of this study. We are aware that the study group is too small to perform a multivariate statistical analysis and present all aspects of the analgesic effect. It was a non-randomized study.

We are convinced that future studies are needed to better establish the analgesic trajectories in different multimodal analgesia models. 

## 6. Conclusions

To conclude, this study indicated that low-opioid anesthesia and non-opioid postoperative analgesia were effective during the first postoperative day in women. However, additional observations from this study indicated that this method was more beneficial in men at the end of postoperative day 1. 

We believe that the results of our study have clinical utility. The results confirmed that the efficacy of general anesthesia with reduced use of opioids and non-opioid analgesia after cholecystectomy is better in the female group, and the chances of achieving better analgesic comfort are higher. In addition, the LOA method provided better analgesia in women at every time interval on the first postoperative day, whereas in men only at the 12–24 h was interval pain intensity significantly lower in the LOA group.

The course of the analgesic scheme based on the NRS in intervals in the group of women anesthetized with LOA formed a negative slope and shows a systematic decrease in pain intensity in subsequent periods of observation. In the group of men, pain intensity was comparable in time 0–2 and 2–6, and in subsequent periods of observation it decreased. The analgesic curve finally formed a negative slope at the end of first postoperative day. Therefore, our results confirmed that the LOA method is beneficial in the group of women.

Thus, we must reject the hypothesis of comparable analgesic quality for LOA and OBA between the male and female, as proposed at the beginning of this study. Opioid analgesia does not provide proper analgesia. The model combining low-opioid anesthesia and non-opioid postoperative analgesia presents a favorable therapeutic effect for women.

## Figures and Tables

**Figure 1 jcm-14-02163-f001:**
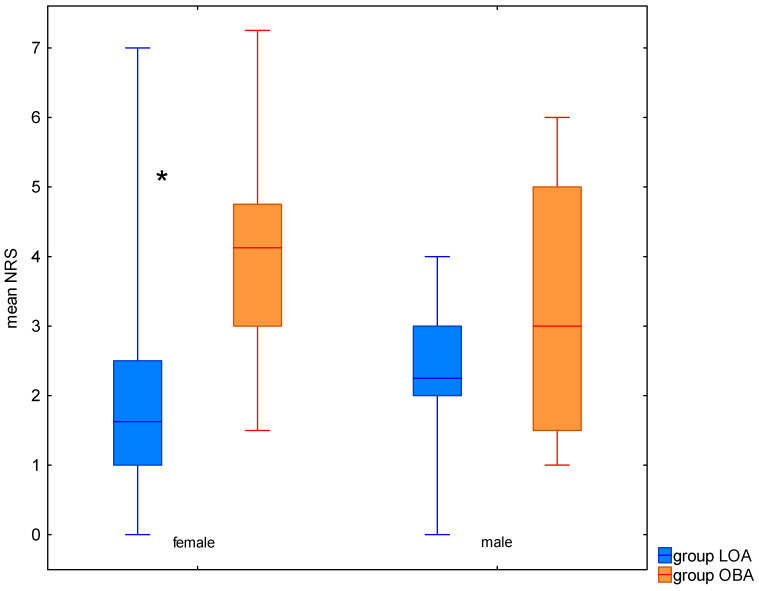
The mean postoperative pain in women and men in LOA and OBA. Median values, minimum–maximum range, and 25–75th percentile are presented. *—statistical significance with *p* < 0.05.

**Figure 2 jcm-14-02163-f002:**
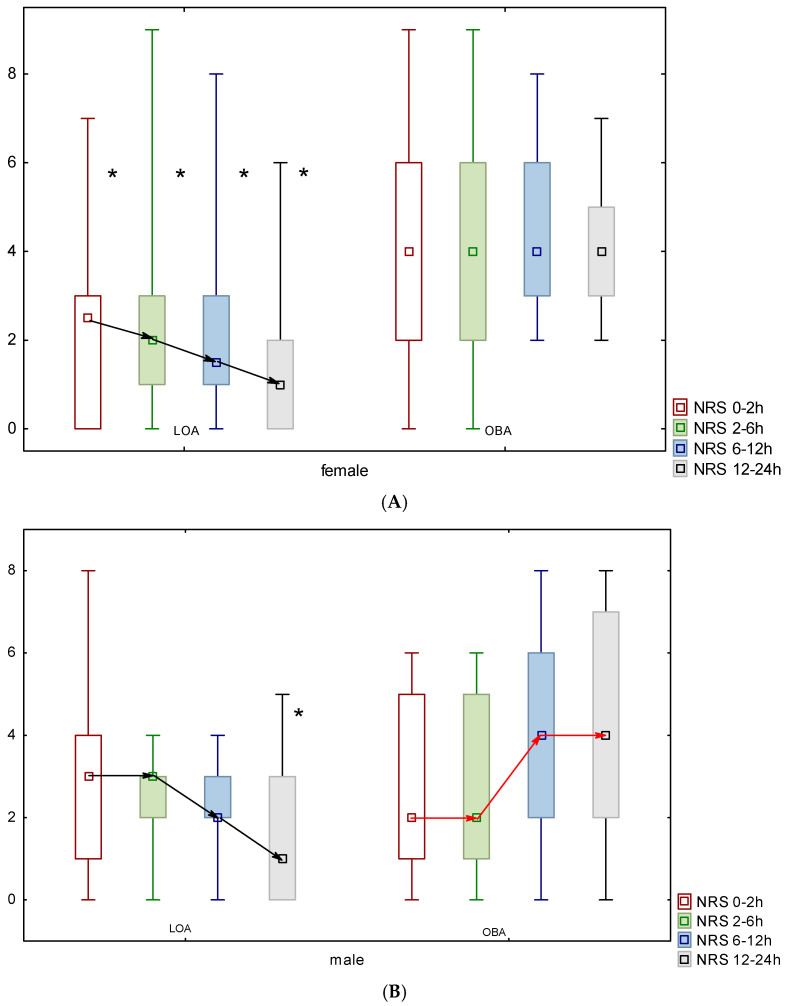
Trajectories of postoperative pain in women (**A**) and men (**B**) in LOA and OBA. Black arrows present negative slope. Red arrows present positive slope. Median values, minimum–maximum range, and 25–75th percentile are presented. *—statistical significance with *p* < 0.05.

**Table 1 jcm-14-02163-t001:** The scheme of LOA and OBA.

	Premedication	Preoperatively	Induction Phase	Maintenance and Recovery Phase of Anesthesia	Postoperatively
LOA	midazolam 7.5 mg p.o	1 g of paracetamol 2 g of magnesium sulfate 100 mg of lidocaine 50 mg of ketamine	0.1 mg of fentanyl 2 mg/kg of propofol 0.15 mg of cisatracurium	infusion of lignocaine (2 mg mL^−1^ solution), at a dose of 1.5–3.0 mg kg^−1^ h^−1^ ketamine (1 mg mL^−1^ solution), at a dose of 0.125–0.25 mg kg^−1^ h^−1^ ketamine infusion was stopped 10 min before the end of surgery lignocaine infusion was terminated at the end of anesthesia 2 g of metamizole 30 min before awakening	1 g of paracetamol 1 g of metamizole intravenously every 6 h
OBA	midazolam 7.5 mg p.o	-	0.1 mg of fentanyl 2 mg/kg of propofol 0.15 mg of cisatracurium	repeated doses of fentanyl of 0.1 mg 2 g of metamizole 30 min before awakening	1 g of paracetamol 1 g of metamizole intravenously every 6 h

**Table 2 jcm-14-02163-t002:** Characteristic of LOA and OBA Groups. The median, minimum, and maximum values are presented. *—statistical significance with *p* < 0.05 compared women in LOA and OBA.

	Low-Opioid Anesthesia (LOA)		Opioid-Based Anesthesia (OBA)		*p*-Value
	Women n = 26	Men n = 13	Women n = 26	Men n = 11	
age	63 (26–79)	50 (40–85)	53.5 (27–81)	65 (31–73)	NS
BMI	28.5 (18–38)	26 (22–34)	27.5 (19–41)	29 (23–38)	NS
BSA	1.84 (1.56–2.21)	1.82 (1.7–2.07)	1.81 (1.54–2.05)	2.04 (1.85–2.35)	NS
Duration of anesthesia (min)	53.5 (30–210)	50 (30–90)	57.5 (30–110)	55 (40–105)	NS
Duration of operation (min)	40 (20–192)	35 (20–75)	40 (25–95)	48 (30–90)	NS
Mean NRS	1.62 * (0–7)	2.25 (0–4)	4.0 (1–8)	2.0 (1–8)	*p* < 0.05
Total dose of fentanyl (mg)	0.1	0.1	0.25 (0.15–0.4)	0.2 (0.2–0.4)	NS

**Table 3 jcm-14-02163-t003:** NRS pain intensity values in time intervals in women and men in OBA and LOA.

	Low-Opioid Anesthesia		Opioid-Based Anesthesia	
	Women n = 26	Men n = 13	Women n = 26	Men n = 13
0–2 h	2.5 * (0–7)	3 (0–8)	4.0 (0–9)	2.0 (0–6)
2–6 h	2.0 * (0–9)	3.0 (0–4)	4.0 (0–9)	2.0 (0–6)
6–12 h	1.5 * (0–8)	2.0 (0–4)	4.0 (2–8)	4.0 (0–8)
12–24h	1.0 * (0–6)	1 ** (0–5)	4.0 (2–7)	4.0 (0–8)

Median values and minimum–maximum ranges are presented. *—statistical significance with *p* < 0.05 compared women in LOA and OBA; **—statistical significance with *p* < 0.05 compared men in LOA and OBA.

## Data Availability

The data are available in the Medical University of Bialystok.

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
