# Peer review of "Sex-Related Analgesic Effects of Opioid-Based Anesthesia and Low-Opioid Anesthesia with Non-Opioid Postoperative Analgesia—A Clinical Observational Study"

_jcm, 2025, doi:10.3390/jcm14072163_

Round 1

Reviewer 1 Report

Comments and Suggestions for Authors

This clinical study is interesting; however, it leaves many doubts about the study design and, in general, about the methodology, since it is explained in a very limited way. For this reason, the following concerns arise that the authors must address in order to improve the study.
- The title of the study does not allow us to know what type of study was carried out. For this reason, we suggest that the authors add to the title whether this study was a clinical trial, a cohort study, or a case-control study.
- Were the study treatments assigned using a randomization method? If so, explain the details in the methodology.
- Was this study blinded? Who was blinded? Explain this in the methodology.
- It is suggested that the calculation of the sample size be reviewed and explained in greater detail.
- In the statistical analysis, it is detailed that the Shapiro Wilks test was used to determine the normality of the data. However, it is not clear which test was used when the data were normal or when the data were not normal. Correct this in the statistical analysis.
- The statistical analysis is confusing and even more confusing to the reader when using the Wilcoxon test for even measurements. So, the obligatory question would be, was the design parallel or was it a cross-over study?
- It is recommended to review the statistical analysis in detail, it is confusing.
- The values ​​of the pain measurements could be graphed in lines, in one graph for women and in another for men. In such a way that the lines allow to clearly observe the differences, if there were any.
- What is the purpose of the ROC curves in this study? It is confusing. These types of curves are made in studies for the development or comparison of diagnostic tests. A different design.
- It may also be interesting to do an analysis without treatment subgroups. Only comparing both treatments. Because having the same number of men and women in the treatment groups we could assume that sex does not influence. Contrary to what the title says, for this reason it would be very interesting to rethink the arguments if necessary.

Author Response

Dear Reviewer,

          Thank you very much for your valuable comments. Based on the review, we have made the following modifications and additions:

-as suggested in the title, we have added a description of the type of study and the title is as follows: Sex-related analgesic effects of Opioid-based anesthesia and Low-opioid anesthesia with non-opioid postoperative analgesia -  a clinical observational study.

-in the Methodology section: we have added additions about the recruitment process and described the criteria for allocation to LOA and OBA

This was non-blinded cohort study and all patients were informed of the purpose of the intended study and how it was to be conducted, and they provided written consent.

Both methods of general anesthesia were considered to be equivalent in analgesic potency and we initially recruited 40 patients to the LOA group, and then recruited 40 patients to the OBA group. The inclusion criteria for the LOA and OBA groups were the same as for recruitment in the study. The study assumption was that both methods in patients with preoperative state of ASA 1 and 2 have no contraindications to anesthesia with these methods.

- in the Statistical analysis: details of stastistical analysis were modified:

                  For normal distribution values, parametric test to compare independent variables and independent groups were used; ROC analysis was removed

- we noticed details about statistical  methods in the Limitations:

                   Secondary, we checked the sample size calculation post-hoc based on the mean NRS value, with an alpha of 0.05 and a power goal of 0.8. The method of power calculation done a priori may improve statistical significance of the study. We are aware that the study group is too small to perform a multivariate statistical analysis and present all aspects of the analgesic effect.

-in the Results section, we have improved the graphic form of the Figures, presentation of the analgesic trajectory is more visible, we have maintained the Table using the publisher's formatting

- we have removed the ROC analysis - also due to the comments of Reviewer2

- we have added several comments in the Conclusions section

We believe that the results of our study have clinical utility. The results confirmed that the efficacy of general anaesthesia with reduced use of opioids and non-opioid analgesia after cholecystectomy is better  in the female group, and the chances of achieving better analgesic comfort are higher. In addition, the LOA method provided better analgesia in women at every time interval on the first postoperative day, whereas in men only at the 12-24 hour interval pain intensity was significantly lower in the LOA group.

The course of the analgesic scheme based on NRS in intervals in the group of women anesthetized with LOA formed a negative slope and shows a systematic decrease in pain intensity in subsequent periods of observation. In the group of men, pain intensity was comparable in time 0-2 and 2-6, and in subsequent periods of observation it decreased. The analgesic curve finally formed a negative slope at the end of 1 postoperative day. Therefore, our results confirmed that the LOA method is beneficial in the group of women.

We hope that the modifications and additions made meet the reviewer's expectations.

Reviewer 2 Report

Comments and Suggestions for Authors

This manuscript presents a valuable contribution to the field of anesthesiology by investigating sex-related differences in analgesic effects between low-opioid anesthesia (LOA) and opioid-based anesthesia (OBA) in laparoscopic cholecystectomy patients. The study addresses an important clinical question with practical implications for perioperative pain management.

Strengths:

  1. Clear research objectives and well-defined primary/secondary endpoints
  2. Novel focus on sex-specific responses to different anesthetic approaches
  3. Systematic methodology with appropriate statistical analysis
  4. Clinically relevant findings that could influence anesthetic practice
  5. Comprehensive literature review and discussion

Major Comments:

Introduction

1. "...decreasing the need for additional sedative, hypnotic, and volatile agents; the hemodynamic response for intubation and extubation; and the surgical stress response." Authors should also add that current guidelines emphasize the importance of comprehensive preoperative assessment in optimizing anesthetic management and outcomes (Lamperti et al., EJA 2025)

Methodology:

  1. The sample size appears relatively small, especially when subdivided by sex. A power calculation was performed post-hoc; it would have been preferable to have this done a priori.
  2. The randomization process isn't clearly described - this should be detailed in the methods section.
  3. The criteria for patient allocation to LOA vs OBA groups should be explained.

Results:

  1. The presentation of pain trajectories could be enhanced with more detailed statistical analysis of the slopes.
  2. The ROC analysis results require more context regarding clinical significance of the AUC values, which appear relatively low (0.53-0.55).

Discussion:

  1. The implications of the findings for clinical practice could be more thoroughly explored.
  2. The limitations section should address potential confounding factors like age and BMI differences between groups.

Minor Comments:

  1. Tables and Figures:
  • Figure 1 needs clearer labeling of axes
  • Table 2 formatting could be improved for better readability
  • Statistical significance indicators should be consistent across all figures
  1. Technical Issues:
  • Several typographical errors need correction
  • Some references are not properly formatted
  • Line numbers occasionally misalign with text

Author Response

Dear Reviewer,

          Thank you very much for your valuable comments. Based on the review, we have made the following modifications and additions:

 - in the Introduction section, we have referred to the current recommendations of Lamperti et al, EJA 2025

           Opioids are primary and multipotent antinociceptive agents that affect all points of the anesthesia triangle with a synergistic pharmacokinetics effect with intravenous and volatile anesthetics as well as benzenediazepines. The pharmacological effects of opioiods reduce the adverse reaction to the various anesthetic activities, including tracheal intubation, and reduce the intensity of surgical stress. While the mechanisms of opioids affect the basic elements of general anesthesia, they are not ideal substances in terms of analgesia and safety due to their side effects, requiring constant efforts to limit their use. Deep general anesthesia based on the use of opioid analgesia is contraindicated in advanced cardiovascular disease and the use of this group of drugs should be considered.

- in the Methodology section: we have added additions about the recruitment process and described the criteria for allocation to LOA and OBA

This was non-blinded cohort study and all patients were informed of the purpose of the intended study and how it was to be conducted, and they provided written consent.

Both methods of general anesthesia were considered to be equivalent in analgesic potency and we initially recruited 40 patients to the LOA group, and then recruited 40 patients to the OBA group. The inclusion criteria for the LOA and OBA groups were the same as for recruitment in the study. The study assumption was that both methods in patients with preoperative state of ASA 1 and 2 have no contraindications to anesthesia with these methods.

 - in the Results section, we have improved the graphic form of the Figures, presentation of the analgesic trajectory is more visible, we have maintained the Table using the publisher's formatting. The data from Figure 2 are supplemented with values presented in the Table 3.

- we have removed the ROC analysis - also due to the comments of Reviewer 2

- the low AUC value disqualifies it from being presented as a significant result

- Discussion and Conclusions were changed so that the parts concerning clinical aspects are the key part

We believe that the results of our study have clinical utility. The results confirmed that the efficacy of general anaesthesia with reduced use of opioids and non-opioid analgesia after cholecystectomy is better  in the female group, and the chances of achieving better analgesic comfort are higher. In addition, the LOA method provided better analgesia in women at every time interval on the first postoperative day, whereas in men only at the 12-24 hour interval pain intensity was significantly lower in the LOA group.

The course of the analgesic scheme based on NRS in intervals in the group of women anesthetized with LOA formed a negative slope and shows a systematic decrease in pain intensity in subsequent periods of observation. In the group of men, pain intensity was comparable in time 0-2 and 2-6, and in subsequent periods of observation it decreased. The analgesic curve finally formed a negative slope at the end of 1 postoperative day. Therefore, our results confirmed that the LOA method is beneficial in the group of women.

 - we have added several comments in the Limitations section

Secondary, we checked the sample size calculation post-hoc based on the mean NRS value, with an alpha of 0.05 and a power goal of 0.8. The method of power calculation done a priori may improve statistical significance of the study. We are aware that the study group is too small to perform a multivariate statistical analysis and present all aspects of the analgesic effect.

-we checked the References.

We hope that the modifications and additions made meet the reviewer's expectations.

Round 2

Reviewer 1 Report

Comments and Suggestions for Authors

The authors have addressed the comments made and a substantial change is seen in the manuscript. However, the statistics are still incorrect, since by designating the study as a cohort study we can assume that the study was parallel. However, the authors still use statistical tests for repeated measures, such as the Wilcoxon test. This test does not need to carry out the description of a paired test, it is used for repeated samples. We recommend that the authors perform the Mann-Whitney U test.

Author Response

Dear Reviewer,

 Thank you very much for cooperation and opinions.

 We changed the  description of statistical analysis as suggested. 

We hope that    improved the manuscript and it will be proper for publication.

 Kind regards

 Urszula Kosciuczuk
